# SOCIALMIRROR: RECONSTRUCTING 3D HUMAN INTERACTION BEHAVIORS FROM MONOCULAR VIDEOS

## ABSTRACT

Accurately reconstructing human behavior in close-interaction scenarios is crucial for enabling realistic virtual interactions in augmented reality, precise motion analysis in sports, and natural collaborative behavior in human-robot tasks. Reliable reconstruction in these contexts significantly enhances the realism and effectiveness of AI-driven interactive applications. However, human reconstruction from monocular videos in close-interaction scenarios remains challenging due to severe mutual occlusions, leading local motion ambiguity, disrupted temporal continuity and spatial relationship error. In this paper, we propose SocialMirror, a diffusion-based framework that integrates semantic and geometric cues to effectively address these issues. Specifically, we first leverage high-level interaction descriptions generated by a vision-language model to guide a semantic-guided motion infiller, hallucinating occluded bodies and resolving local pose ambiguities. Next, we propose a sequence-level temporal refiner that enforces smooth, jitter-free motions, while incorporating geometric constraints during sampling to ensure plausible contact and spatial relationships. Evaluations on multiple interaction benchmarks show that SocialMirror achieves state-of-the-art performance in reconstructing interactive human meshes, demonstrating strong generalization across unseen datasets and in-the-wild scenarios. The code will be released upon publication.

## 1 INTRODUCTION

Human reconstruction, which recovers the 3D geometry and motion of human bodies from visual inputs, is a fundamental computer vision task, which has extensive applications in fields such as augmented reality (Urgo et al. (2024)), sports analysis (Fukushima et al. (2024); Xi et al. (2024)), and film animation. Close human interactions(Huang et al. (2024); Müller et al. (2024)), including social and competitive behaviors, are particularly critical in these contexts. The interaction further plays a crucial role in robotics applications, where collaborative tasks require seamless human-robot interaction. Accurately modeling human behavior in such interactions allows robots to engage in more natural, human-like collaborations, aligning with human preferences and enhancing the effectiveness of AI in interactive tasks.

Previous monocular human reconstruction works (Kanazawa et al. (2018); Bogo et al. (2016)) primarily target single-person scenarios. These methods typically focus on accurate pose estimation (Li et al. (2021); Rempe et al. (2021)), shape reconstruction fidelity (Goel et al. (2023); Pavlakos et al. (2019); Xu et al. (2020)), or temporal smoothness across frames (Kocabas et al. (2020); Zheng et al. (2021); Zeng et al. (2022)). However, limiting reconstruction to single-person scenarios restricts applicability in real-world multi-person interactive settings. A few works(Huang et al. (2023); Lu et al. (2023); Ugrinovic et al. (2024); Sun et al. (2022); Su et al. (2025); Newell et al. (2025); Liu et al. (2025)) have considered reconstructing multi-human poses, employing techniques such as explicit collision avoidance constraints (Ugrinovic et al. (2024)), depth ordering modeling in crowded scenes (Sun et al. (2022); Wen et al. (2023)), utilizing data-driven priors(Zhu et al. (2024); Lu et al. (2023); Rempe et al. (2021); Shi et al. (2023)) or recovery relation reasoning(Huang et al. (2023)). The methods mentioned above typically address multi-person scenarios, where individuals are in the same space but not directly interacting. In contrast, close-interaction scenes often involve heavy occlusions, especially when individuals are physically touching or positioned in tight spaces, which is more relevant for collaborative tasks and robot-human interactions. While some methods(Müller et al. (2024); Huang et al. (2024); Fang et al. (2024)) have used mutual priors to model interactions,

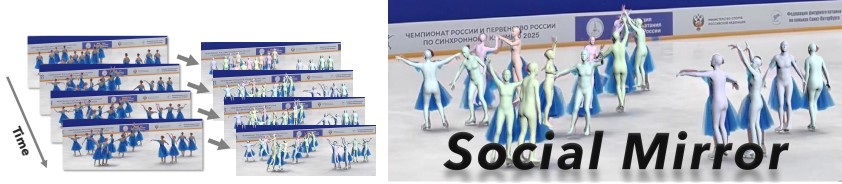

Figure 1: We reconstruct 3D human motion from monocular videos, specifically target on close interaction scenarios. By leveraging both semantic and geometric guidance, SocialMirror resolves ambiguities through infilling and ensuring the spatial relationship.

reconstructing closely interacting humans from monocular videos remains challenging due to the exclusive reliance on image features under severe occlusions. This leads to three critical issues: (1) **Local pose ambiguities** occur occludes another, making it difficult to infer the hidden person's pose, leading to uncertainties. (2) **Temporal inconsistencies** arise when occlusions disrupt tracking continuity, resulting in unrealistic motion, such as sudden changes in pose. (3) **Spatial relationship errors** happen when image features alone fail to capture the dynamic, complex interactions between individuals in close proximity, causing inaccuracies in contact areas.

To address the aforementioned issues, we observe that human interactions are inherently intentional, suggesting the necessity of incorporating semantic context to infer motion and spatial relationships. Additionally, as interactions naturally occur within 3D space, enforcing geometric constraints is crucial for achieving physically plausible reconstructions. Motivated by these insights, we propose **SocialMirror**, a Semantic and Geometric guided framework for Interactive Human Mesh Reconstruction from monocular video, shown in Figure 1. First, we introduce *Semantic-Guided Motion Infiller* which incorporates textual semantic guidance alongside visual features to recover motion in occluded regions, effectively addressing severe occlusions and image-feature degradation. Specifically, a Vision-Language Model (VLM)(Bai et al. (2025); Achiam et al. (2023)) Annotator first generates textual descriptions of human interactions, providing essential semantic context and temporal cues. These annotations, along with image features from visible body parts extracted by a pre-trained backbone, guide the process to reconstruct infilled motion from invisible regions, ensuring semantically coherent reconstruction despite occlusions. Furthermore, the *Temporal Motion Refiner* module performs sequence-level optimization, merging diffusion-based semantic content with visibly reconstructed sequences. This ensures spatially and temporally coherent reconstructions even when image information is severely degraded or unavailable. To accurately model the complex spatial relationships between closely interacting humans, the *Geometry Optimizer* explicitly captures geometric constraints based on 3D joint positions. This geometric supervision refines reconstructed motion sequences, effectively resolving spatial ambiguities and improving realism in contact regions.

Experimental evaluations demonstrate that the proposed method achieves superior reconstruction accuracy on human interaction datasets with particular advantages in capturing interpersonal spatial relationships and interaction plausibility. To summarize, our work makes the following contributions: (1) We introduce semantic information into monocular video-based human mesh reconstruction via a diffusion-based framework. Semantic guidance enables the network to infer plausible poses in occluded regions, effectively resolving ambiguities through motion infilling. In addition, semantic context provides essential temporal cues and contact relationships, enhancing reconstruction accuracy in closely interacting regions. (2) The proposed method incorporate temporal refinement and 3D geometric guidance, ensuring temporal consistency, spatial relationship and geometric plausibility. (3) Experimental validation confirms our method achieves superior reconstruction quality in monocular interactive human scenarios. Notably, the approach demonstrates generalization capabilities across unseen datasets and in-the-wild scenarios.

## 2 RELATED WORK

### 2.1 HUMAN RECONSTRUCTION

Building on advancements in single-person 3D reconstruction(Kanazawa et al. (2018); Bogo et al. (2016)), recent approaches have increasingly focused on joint reconstruction of multiple individuals from monocular images. Prior works(Fieraru et al. (2021); Jiang et al. (2020); Zanfir et al. (2018); Sun et al. (2021; 2022); Fieraru et al. (2020); Li et al. (2022)) have focused on improving human relative position and depth estimation, using strategies including depth ordering losses(Fieraru et al.

(2021); Jiang et al. (2020)), collision constraints(Zanfir et al. (2018); Sun et al. (2021)), and bird's-eye view depth reasoning(Sun et al. (2022)). However, these methods still struggle with occlusions and modeling inter-person relationships. To address these issues, some works enhance feature extraction under occlusion(Kocabas et al. (2021); Baradel* et al. (2024)), incorporate pose priors(Zhu et al. (2024); Lu et al. (2023); Rempe et al. (2021); Shi et al. (2023)), and use contextual motion completion frameworks(Yuan et al. (2022)) for inferring. Additionally, GroupRec(Huang et al. (2023)) improves human mesh recovery through relational reasoning. However, these approaches fail to capture the complex interpersonal interactions and accurate contact in close-range scenarios. Only a few studies(Müller et al. (2024); Ugrinovic et al. (2024); Huang et al. (2024); Fang et al. (2024)) explicitly address close interactions which involve more intimate contact and heavy occlusions. BUDDI(Müller et al. (2024)) introduces a diffusion-based prior but is limited to static images. MultiPhys(Ugrinovic et al. (2024)) resolves mesh interpenetration using a physics engine, while CloseInt(Huang et al. (2024)) applies mutual attention modules for iterative refinement from monocular video. However, all of these methods overlook the semantic context inherent in close human interactions and still face challenges with visual ambiguities.

## 2.2 HUMAN MOTION GENERATION

Human motion generation has progressed from single-person motion generation(Tevet et al. (2022); Chen et al. (2023); Jiang et al. (2024)) to more complex human-human interaction generation. Approaches include response synthesis (Chopin et al. (2023); Ghosh et al. (2024); Liu et al. (2023); Xu et al. (2024)), where motion is generated in response to an actor's movements, and interaction generation(Liang et al. (2024); Tanaka & Fujiwara (2023)), which generates motions for all interacting individuals simultaneously. The methods mentioned above primarily focus on motion generation without explicit control. However, control-based approaches, such as motion completion(Choi et al. (2021); Chung et al. (2022); Zhao et al. (2024); Tevet et al. (2022)) and trajectory or joint-based(Wan et al. (2024)) control frameworks, have introduced greater control, improving the coherence and diversity of the generated outputs. OmniControl (Xie et al. (2023)) and InterControl (Wang et al. (2023)) integrate ControlNet(Zhang et al. (2023b)) and enforce joint constraints and physical plausibility, ensuring more accurate and realistic motion generation. Control-based diffusion is especially beneficial for human reconstruction tasks, where accurate alignment with input images and the ability to handle occlusions or depth ambiguities are crucial for maintaining both temporal and spatial consistency in the generated motions.

## 2.3 LLM IN POSE EATIMATION

Large language models (LLMs)(Achiam et al. (2023)) are known for their strong generalization capabilities, particularly in introducing semantic information. Their semantic flexibility and generalization have been proven effective in pose estimation tasks. Xiao et al. (2025) integrates Swin-Base image features with CLIP's text-image embeddings, creating multimodal conditional inputs that improve pose understanding. Wang et al. (2025) uses SHAPY(Choutas et al. (2022)) to generate body shape description texts and fuses the encoded text prompts with other features to improve monocular body shape estimation accuracy. Subramanian et al. (2024) leverages a LLM to generate contact constraints between body parts, transforming these into a loss function to enforce physically consistent predictions for both self-contact and interpersonal interactions. Xu et al. (2025) uses a vision-language model to extract detailed descriptions of body part interactions, which are then used as multimodal feedback to refine initial pose estimates. Building on this, we extend these techniques to human reconstruction from monocular videos in close-interaction scenarios, where we combine visual and textual cues with VLM(Bai et al. (2025)) and LLM to generate more accurate and semantically coherent interactions. We incorporate temporal contact labels and refine the reconstruction process to ensure not only spatial consistency but also temporal continuity and geometric constraints.

## 3 METHOD

The aim of this work is to reconstruct human close interactions from monocular videos. We introduce SocialMirror, a semantic and geometry-guided diffusion-based framework for interactive human mesh reconstruction, as shown in Figure. 2. Specifically, we extract textual descriptions and labels with temporal and close-contact information from a Vision-Language Model (VLM) and integrate

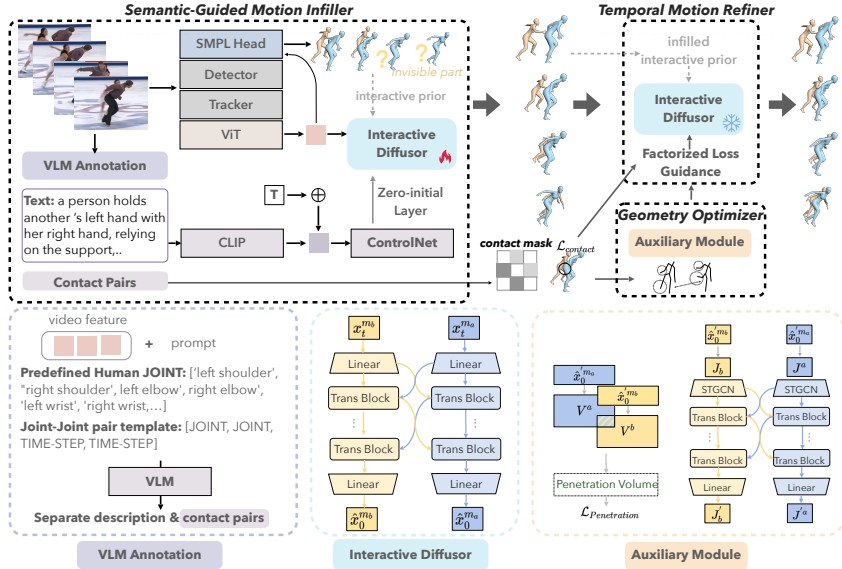

Figure 2: The framework of SocialMirror, which integrates semantic guidance from vision-language annotations and further refine the result with geometric constraints. *Trans Block* refers to the transformer block.

these semantic features with visual data in Semantic-Guided Motion Infiller 3.1, which compensates for visual feature degradation when severe occlusions and mitigates local pose ambiguities. The Geometry Optimizer 3.2 uses an auxiliary model to optimize 3D joint positions, generating geometric guidance signals to better model spatial relationships. The Temporal Motion Refiner 3.3 refines the reconstruction results based on these geometric signals, ensuring temporal consistency.

## 3.1 SEMANTIC-GUIDED MOTION INFILLER

In multi-person close-interactive scenarios, severe partial occlusion frequently occurs, causing certain individuals to become visually obscured. Under such challenging conditions, existing reconstruction methods(Müller et al. (2024); Huang et al. (2024)) typically struggle due to the lack of reliable visual features from occluded subjects. Nevertheless, human observers consistently maintain perceptual coherence in these scenarios by effectively utilizing semantic information: even when visual details are obscured, contextual cues regarding interactive dynamics allow humans to infer plausible states of hidden regions via spatial and temporal reasoning. Inspired by this, the motion reconstruction models should leverage high-level semantic understanding rather than relying solely on pixel-level visual restoration. Consequently, our aim is to enable models to learn semantic-to-motion mappings, empowering the model to inpaint invisible regions through available visual cues and inferred interaction semantics.

**VLM Annotator** Large language models offer strong generalization capabilities and rich semantic information. Leveraging inputs such as detailed background scene data, human joint information, and predefined instructions, we use a vision-language model to generate semantic captions for the interacted motion of two people. These captions are then converted into single-person descriptions through prompt engineering. Additionally, we introduce sequential and spatial-level contact labels to guide the language model in modeling interactions, which are used in the Temporal Motion Refiner and Geometry Optimizer. We pre-calculate the minimum distance between joints of the individuals and label pairs with a distance below a threshold as contact. Each contact pair is then formatted as (JOINT, JOINT, BEGIN-CONTACT-TIME-STEP, END-CONTACT-TIME-STEP) for further processing. We fine-tune the VLM to enable the model to infer contact labels. Details of the template design are provided in the Appendix.

**Feature Extractor** The input consists of sequential images, with the frames length equal to $\mathbf{L}$, and the output is the SMPL parameter for each person, representing their motion, which includes the local pose $\theta \in \mathbb{R}^{21 \times 3}$, shape $\beta \in \mathbb{R}^{10}$, rotation $\phi \in \mathbb{R}^3$, and translation $\tau \in \mathbb{R}^3$. The parameter for a single person is defined as $\mathbf{x} = \{\phi, \theta, \beta, \tau\}$. The reconstructed results must primarily adhere to visual evidence. For the visible parts, we leverage the existing HMR framework(Ge et al. (2021);

Lyskov (2024); Goel et al. (2023)) to obtain initial estimation and image features. We first employ off-the-shelf human detection and tracking methods(Lyskov (2024); Cheng & Schwing (2022); Zhang et al. (2023a); Jocher et al. (2023); Ge et al. (2021)) to acquire the bounding boxes of individuals in images, and then a Vision transformer (ViT) (Goel et al. (2023)) pretrained on extensive datasets serves as the backbone network to extract image features $F_{img}$ within these bounding boxes. We further apply a motion head with sequential MLP layers to obtain SMPL Token from $F_{img}$, and we derive the initial coarse estimates $\mathbf{x} = \{x_a, x_b\}$. For the interactive descriptions generated by the VLM Annotator, we use CLIP(Radford et al. (2021)) as the text encoder to obtain textual features $F_{\text{text}}$. We incorporate semantic cues to facilitate complete reconstruction for occluded body regions.

**Interactive Diffuser** The Interactive Diffuser integrates visual features from observable body regions with textual semantic guidance to generate interactive motions. Recent advancements in controllable diffusion-based generation(Xie et al. (2023); Wan et al. (2024); Zhang et al. (2023b)) are ideal for our task: visible body regions require strict adherence to input images, while occluded regions need context-aware completion. Unlike traditional diffusion models that start from pure noise, we generate from the coarse motion $\mathbf{x}$ with Distribution Adaptation (Huang et al. (2024)), which ensures output consistency with observed human poses and preserves key pose features in the generated results.

Following prior methods (Wang et al. (2023); Xie et al. (2023)), the Interactive Diffuser takes the interactive individual motions, extracted from SMPL head, $x_a^t$ and $x_b^t$, as denoising inputs. It then produces the corresponding denoised motions $\hat{x}_a^0$ and $\hat{x}_b^0$, conditioned on the diffusion timestep $t$ and image features $F_{img}$. The textual descriptions serve as auxiliary guidance through a zero-initialized layer, similar to ControlNet(Zhang et al. (2023b)). Human interactions inherently involve mutual influence between individuals' movements. To model this, we adopt a dual-branch structure with cross-attention mechanisms (Liang et al. (2024)), where each branch handles motion reconstruction for one individual while maintaining shared weights and bidirectional information exchange. This configuration effectively captures the reciprocal nature of interactive motions. For more details of model architecture, we introduce in appendix.

**Model Training** We optimize through following objectives fuction: $\mathcal{L} = \mathcal{L}_{\text{reproj}} + \mathcal{L}_{\text{smpl}} + \mathcal{L}_{\text{joint}} + \mathcal{L}_{\text{vel}} + \mathcal{L}_{\text{int}} + \mathcal{L}_{\text{pen}}$, where $\mathcal{L}_{\text{reproj}} = \|\Pi(J + \tau) - \hat{J_{2D}}\|_2^2$, measuring the projected 3D joints and the 2D ground truth poses, $J \in \mathcal{R}^{21 \times 3}$ is the 3D joint derived from smpl parameters. $\mathcal{L}_{\text{smpl}}, \mathcal{L}_{\text{joint}}, \mathcal{L}_{\text{vel}}$ are the $\mathcal{L}_2$ between predicted and target of shape parameters, 3D joint positions and joint velocities. $\mathcal{L}_{\text{int}} = \||J_a - J_b| - |\hat{J}_a - \hat{J}_b|\|_2^2$ supervise the relative distance between two invdividuls. For Penetration Loss, we first detect the set of colliding triangles using bounding volume hierarchies (BVH) (Karras (2012)), then calculate Penetration Loss by:

$$\mathcal{L}_{pen} = \sum_{(f_a, f_b) \in \mathcal{C}} \left\{ \sum_{v_a \inf_a} \|-\Psi_{f_b}(v_a) n_a\|^2 + \sum_{v_b \inf_b} \|-\Psi_{f_a}(v_b) n_b\|^2 \right\} \tag{1}$$

Where $f_a, f_b$ are two colliding triangles in the detected colliding triangles $\mathcal{C}.v$ and $n$ are vertex position and normal, respectively, and $\Psi(\cdot)$ is the distance field.

## 3.2 GEOMETRY OPTIMIZER

In the prior diffusion module, interaction sequences were modeled as a set of SMPL parameters and position parameters within the camera coordinate system. This modeling approach may diminish the model's capacity to grasp spatial relationships, while explicit modeling of the 3D positions of human joints facilitates the model's learning of relative joint position relationships in two-person interactions. Therefore, for the reconstruction results of the diffusion Model, we apply an auxiliary Module to obtain the 3D joint positions of the two individuals in the interaction sequence, and then further optimize the final motion sequence.

**Auxiliary Module.** The Auxiliary Module adopts the same two-branch mutual attention structure as the diffusion model. The key difference is that the linear layer in the Motion Embedding component is replaced with a Spatial-Temporal Graph Convolutional Network (STGCN) (Yu et al. (2017); Yan et al. (2018)), which models spatio-temporal relationships. Based on the human anatomical structure, nodes in each frame are connected to form spatial edges, while temporal edges link corresponding joints across consecutive time steps. This setup enables the construction of multi-layer spatial-temporal graph convolutions, facilitating the integration of information across both spatial and temporal

dimensions. We convert the joint pair annotations from the VLM Annotator into a contact mask $M \in \mathcal{R}^{K \times L}$, indicating contact joints from the start of one time step to the end, with $M_{k,l} = 1$ if there is contact and $M_{k,l} = 0$ if there is no contact. This model is trained with a composite loss function defined as $\mathcal{L} = \mathcal{L}_{\text{reproj}} + \mathcal{L}_{contact} + \mathcal{L}_{\text{vel}} + \mathcal{L}_{\text{int}}$, where $\mathcal{L}_{contact} = (\alpha M + \mathbf{1}_{K \times L})\mathcal{L}_{\text{joint}}$, which deliberately strengthen the contact positions and 3D geometric information, thereby enhancing the spatial relationships.

## 3.3 TEMPORAL MOTION REFINER

Through the Semantic-Guided Motion Infiller, we generate an interaction motion sequence conditioned on both visual and semantic cues. However, textual guidance may unintentionally alter visible regions, and interpenetration artifacts can still occur due to the use of soft collision penalties. To address this issue, we apply a confidence-based infilling strategy. Given the initial estimation $\mathbf{x}$ from SMPL head, the infilled sequence $\hat{x}^0$, and a confidence mask $\mathcal{C}$, the final motion sequence is obtained as $x' = M \odot \mathbf{x} + (1 - M) \odot \hat{x}^0$, where $M = \mathbf{1}_{\{\mathcal{C} \geq \theta\}}$ is a binary mask derived from the confidence scores $\mathcal{C} \in [0, 1]^T$ with threshold $\theta \in [0, 1]$, and $\odot$ denotes element-wise multiplication. Thus, through this operation, the textual guidance implicitly hallucinates the low-confidence regions, while the high-confidence parts are preserved from the initial estimation. Subsequently, we further optimize the infilled motion sequence with a frozen Interactive Diffuser, which leverages the generative prior of diffusion models to naturally improve temporal coherence and produce smoother transitions across frames. In parallel, it integrates guidance from the Geometry Optimizer to refine spatial relationships, leading to motion sequences that are both geometrically consistent and temporally smooth. Additionally, we introduce a factorized collision loss that enables joint constraints and collision-guided sampling, independently optimizing body shape and joints for efficient convergence.

**Factorized Loss Guidance.** During the sampling process, we incorporate joint and collision guidance signals to improve interaction quality. The joint guidance signal is derived from the joint positions generated by the Geometry Optimizer. For the predicted motion $\hat{\mathbf{x}}_0$ from freezed interactive diffuser, we derive it 3D joint positions $J$ and compute the weight contact loss between $J$ and the guidance signal $J'$ from the Geometry Optimizer as $\mathcal{L}_{\text{contact}}(J, J') = (\alpha M + \mathbf{1}_{K \times L})||J - J'||^2$. The collision guidance signal is based on the interpenetration volume between meshes, which reduces mesh penetration, improving the geometric plausibility of the interaction. We reconstruct meshes for the two individuals from $\hat{x}_0$ and use BVH to calculate their $L_2$ intersection volume differences as $\mathcal{L}_{\text{penetration}}$. The guidance loss is defined as $\mathcal{L}_{\text{guidance}} = \lambda_j \mathcal{L}_{\text{contact}} + \lambda_p \mathcal{L}_{\text{penetration}}$, where $\lambda_j$ and $\lambda_p$ are weighting parameters. Following the methodology in InterControl (Wang et al. (2023)), we perform multiple L-BFGS iterations at each denoising step to update the posterior mean. The optimization process is described as follows: $\mu'_t = \mu_t - \lambda \nabla_{\mu_t} \mathcal{L}_{\text{guidance}}(\mu_t)$, where $\lambda$ denotes the optimization step size.

In addition, joint optimization of heterogeneous parameters (rotation, shape, and translation) with uniform settings leads to suboptimal outcomes. To address this, we introduce a factorized loss guidance approach. Since joint guidance provides limited shape-related information and collision constraints may cause unwanted morphological compression when applied to shape parameters, we decompose the optimization process into two components: rotational parameters $\mu_t^{\text{pose}}$ and translational parameters $\mu_t^{\text{transl}}$ for optimizing separately. Each component undergoes multi-round iterative optimization with L-BFGS optimizer. This factorized approach allows for task-specific optimization, leading to more efficient convergence and more plausible results.

## 4 EXPERIMENT

### 4.1 DATASETS

**Hi4D** (Yin et al. (2023))focuses on close human interaction scenarios, encompassing dynamic interaction types such as hugging, dancing, and athletic movements. It specifically addresses complex poses and prolonged physical contact scenarios, thereby challenging existing methods' capacity to handle occlusions and interactive dynamics. The dataset comprises 20 unique participant pairs, totaling 100 sequences with over 11,000 frames of which more than 6,000 frames contain physical contact. For consistency, we adopt the same train and test split protocol as the baseline.

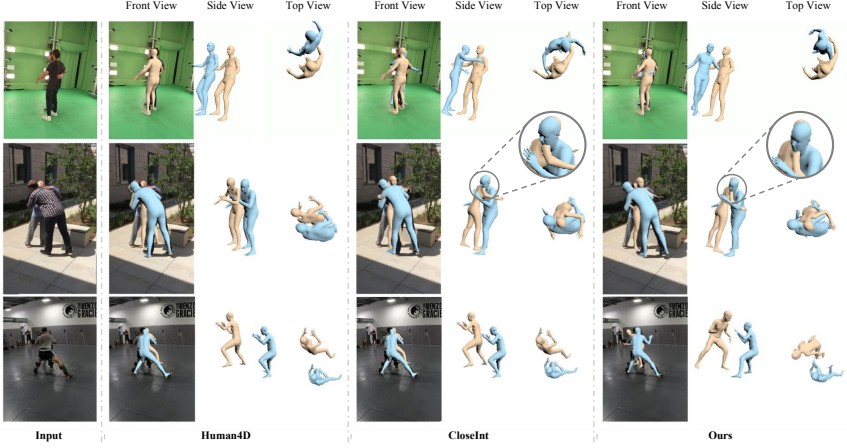

Figure 3: Qualitative comparison results.

Table 1: Comparisons on Hi4D and 3DPW. PA. represents PA-MPJPE. and VPE represents MPVPE.

| Method | Hi4D | | | | | | | | 3DPW | | | | | | | |
|---|---|---|---|---|---|---|---|---|---|---|---|---|---|---|---|---|
| | ↓RE | ↓GE | ↓Int. | ↓Smoothness. | ↓Pen. | ↓MPJPE | ↓PA. | ↓VPE | ↓RE | ↓GE | ↓Int. | ↓Smoothness. | ↓Pen. | ↓MPJPE | ↓PA. | ↓VPE |
| Human4D(Goel et al. (2023)) | - | - | - | - | - | 72.1 | 52.4 | 88.6 | - | - | - | - | - | 72.9 | 49.1 | 107.0 |
| BEV(Sun et al. (2022)) | 210.5 | 223.5 | 131.0 | - | 1953.6 | 91.8 | 59.3 | 101.2 | 235.2 | 291.8 | - | 145.6 | 233.8 | 135.0 | 81.9 | 169.7 |
| GroupRec(Huang et al. (2023)) | 113.2 | 122.3 | 98.8 | - | **1858.4** | 82.4 | 51.6 | 88.6 | 204.6 | 235.2 | - | 110.6 | **100.9** | 73.3 | 48.7 | 109.4 |
| BUDDI(Müller et al. (2024)) | 200.3 | 216.4 | 102.6 | - | 1879.3 | 96.8 | 70.6 | 116.0 | 228.4 | 289.4 | - | 113.1 | 203.5 | 83.6 | 53.6 | 93.8 |
| CloseInt(Huang et al. (2024)) | 99.0 | 114.9 | 81.4 | 4.6 | 3947.6 | 63.1 | **47.5** | **76.4** | 121.1 | 134.0 | 19.9 | 75.6 | 101.6 | 59.0 | 45.3 | 73.2 |
| Ours | **83.6** | **95.2** | **68.5** | **3.5** | 2380.5 | **62.2** | **47.5** | 79.3 | **91.0** | **127.9** | **10.0** | **64.6** | 109.7 | **55.6** | **44.3** | **69.4** |

**3DPW** (Von Marcard et al. (2018)) records human activities in natural environments, encompassing various daily scenarios such as courtyard, downtown, and office. We selected sequences involving two-person interactions from these recordings, resulting in a total of 31 sequences with 12,000 frames.

**Harmony4D** (Khirodkar et al. (2024)) is a multi-view video dataset specialized in in-the-wild close human interactions. Unlike datasets collected in controlled settings with choreographed motions, Harmony4D captures naturally occurring dynamic activities including wrestling, dancing, and mixed martial arts. The dataset contains 208 video sequences captured by over 20 synchronized cameras, yielding 1.66 million images across 5 distinct scenarios involving 24 unique participants. We utilize the test set of this dataset to validate the generalization ability on unseen dataset without training.

Table 2: Comparisons on Harmony4D.

| | ↓RE | ↓GE | ↓Int. | ↓Pen. | ↓MPJPE | ↓PA. | ↓VPE |
|---|---|---|---|---|---|---|---|
| Human4D | - | - | - | - | 108.2 | 60.3 | 131.0 |
| BEV | 365.4 | 716.7 | 360.4 | 484.4 | 111.3 | 78.0 | 144.3 |
| GroupRec | 346.6 | 689.2 | 337.1 | 499.4 | 119.0 | 65.5 | 144.8 |
| BUDDI | 352.3 | 692.3 | 324.1 | **479.3** | 126.4 | 84.0 | 158.7 |
| CloseInt | 202.2 | 446.6 | 255.2 | 488.9 | **103.5** | 47.1 | **114.9** |
| Ours | **198.2** | **411.8** | **245.6** | 482.9 | 104.6 | **45.9** | 117.3 |

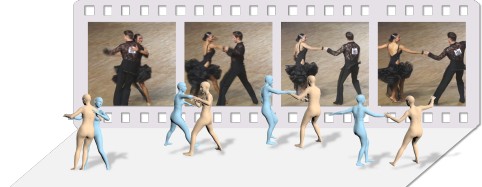

Figure 4: Visulization on in-the-wild video.

## 4.2 EVALUATION METRICS

We adopt the evaluation metrics mainly from CloseInt(Huang et al. (2024)), including Root-Aligned Mean Per Joint Position Error (**MPJPE**) and Procrustes-aligned MPJPE (**PA-MPJPE**) for pose estimation accuracy. Mean Per Vertex Position Error (**MPVPE**) measures mesh reconstruction quality. These metrics primarily evaluate the proposed method's accuracy in single-person pose reconstruction tasks. To further evaluate the network's capacity to model spatial relationships in multi-person interactions, except for **Interaction** defined in CloseInt, we introduce two complementary metrics: Global Mean Per Joint Position Error (**G-MPJPE(GE)**) measuring absolute pose errors across the entire scene, and Relative Mean Per Joint Position Error (**R-MPJPE(RE)**) focusing on inter-person positional relationships, which is defined as Mean Per Joint Position Error after aligning to the first person's root position. It eliminates the effect of global position offset and focuses more on the relative position relationship between the joints. For temporal consistency, we follow MultiPhys (Ugrinovic et al. (2024)) and report **Smoothness**, calculated as the Mean Squared Error between the predicted and ground-truth accelerations of each joint. This metric quantifies the continuity of joint movements across the temporal sequence. For physical plausibility, we report inter-person penetration volume (**Pen.**), quantified by computing the signed distance function (SDF) for each subject and accumulating the penetration depth across intersecting vertices. The Pen. metric represents the average sum of negative SDF values per person over the entire sequence.

Table 3: Ablation studies on the impact of semantic and geometric information on Hi4D.

| Semantic-Guided Motion Infiller | Temporl Refiner - | Temporl Refiner factorized guidance | Temporl Refiner contact mask | Geometry Optimizer | ↓RE | ↓GE | ↓Int. | ↓Smoothness. | ↓MPJPE | ↓PA. | ↓VPE |
|---|---|---|---|---|---|---|---|---|---|---|---|
| | | | | | 100.4 | 119.0 | 90.5 | 4.7 | 62.4 | 47.5 | **78.3** |
| ✓ | | | | | 91.2 | 102.7 | 73.4 | 4.1 | 63.5 | 48.7 | 80.7 |
| ✓ | ✓ | | | | 91.5 | 103.5 | 74.2 | 3.9 | 63.3 | 48.5 | 80.2 |
| ✓ | ✓ | | | ✓ | 91.0 | 102.4 | 73.3 | 4.1 | 63.5 | 48.7 | 80.7 |
| | ✓ | ✓ | | ✓ | 89.3 | 100.6 | 69.2 | 4.0 | 63.2 | 48.5 | 80.3 |
| | ✓ | ✓ | ✓ | ✓ | 88.6 | 98.7 | 68.7 | 3.9 | 63.0 | 48.5 | 80.2 |
| ✓ | ✓ | ✓ | | ✓ | 84.5 | 96.2 | 68.5 | **3.5** | 62.8 | 47.7 | 79.8 |
| ✓ | ✓ | ✓ | ✓ | ✓ | **83.6** | **95.2** | 68.5 | **3.5** | **62.2** | **47.5** | 79.3 |

Table 4: Results under various occlusion severity on Hi4D. Darker colors indicate greater improvements. Improve. means the improvement rate.

| IoU | 0.0 | | | (0, 0.25] | | | (0.25, 0.5] | | | (0.5, 0.75] | | | (0.75, 1.0] | | |
|---|---|---|---|---|---|---|---|---|---|---|---|---|---|---|---|
| | CloseInt | Ours | Improve. | CloseInt | Ours | Improve. | CloseInt | Ours | Improve. | CloseInt | Ours | Improve. | CloseInt | Ours | Improve. |
| ↓RE | 85.4 | 78.3 | 8.3 | 93.3 | 82.3 | 11.8 | 104.1 | 91.4 | 12.2 | 104.3 | 93.8 | 10.1 | 108.4 | 106.7 | 1.6 |
| ↓GE | 99.4 | 89.0 | 10.5 | 112.4 | 96.3 | 14.3 | 120.4 | 101.2 | 16.0 | 118.3 | 105.3 | 11.0 | 125.6 | 117.7 | 6.3 |
| ↓Int. | 67.8 | 62.6 | 7.7 | 87.2 | 68.6 | 21.3 | 98.2 | 79.3 | 19.3 | 93.3 | 77.5 | 16.9 | 93.3 | 82.4 | 11.7 |
| ↓Pen. | 15.6 | 2.8 | 82.1 | 139.8 | 48.7 | 65.16 | 1232.2 | 292.1 | 76.3 | 2690.6 | 406.3 | 84.9 | 4218.5 | 111.2 | 97.4 |
| ↓MPJPE | 40.9 | 44.7 | -9.3 | 49.2 | 51.3 | -4.3 | 64.2 | 65.7 | -2.3 | 74.6 | 72.6 | 2.7 | 84.1 | 88.1 | -4.8 |
| ↓PA. | 31.1 | 35.2 | -13.2 | 37.6 | 40.1 | -6.7 | 49.3 | 51.3 | -4.1 | 55.6 | 54.9 | 1.3 | 58.6 | 60.6 | -3.4 |
| ↓VPE | 52.8 | 58.4 | -10.6 | 64.1 | 67.9 | -5.9 | 80.6 | 83.5 | -3.6 | 91.8 | 90.4 | 1.53 | 100.4 | 108.1 | -7.7 |

## 4.3 MAIN RESULTS

**Results on Hi4D and 3DPW.** We compare our method with several state-of-the-art baseline methods on Hi4D(Yin et al. (2023)) and 3DPW(Von Marcard et al. (2018)), most values are reported in the original paper, and we reproduce the results for interaction metric evaluation. A dash (-) indicates that some results are either not reported or unavailable. While Human4D (Goel et al. (2023)) achieves promising results on single-person metrics, it does not account for the mutual relationships of interacting individuals and fail to capture the spatial dependencies between different subjects. BEV(Sun et al. (2022)) and GroupRec(Huang et al. (2023)) explicitly consider the depth relationships among humans to address the depth ambiguity of human positions in monocular multi-person reconstruction tasks, while they struggle to handle the complex interaction patterns in scenarios with close human interactions. BUDDI(Müller et al. (2024)) and CloseInt(Huang et al. (2024)) share the most similar setting with our method, focusing on monocular two-person reconstruction under close multi-person interaction. BUDDI uses a Generative Proxemics model to align meshes with the initial estimate and detected keypoints. The quality of its results relies on the accuracy of the keypoints, which can be unreliable or missing when humans are heavily occluded. Additionally, BUDDI lacks temporal modeling for handling dynamic interactions over time. CloseInt employs a two-person interaction prior, which also relies on the precondition that the movements of both individuals can be roughly reconstructed. Therefore, they struggle to handle occlusion in monocular videos, resulting in poor interaction relationships. In contrast to these methods, our approach introduces semantic information to infill occluded body parts, operating without reliance on 2D keypoint detection or flawed image features. We achieves 4.2% and 18.3% improvements in RE and Int. compared with the latest SOTA on Hi4D. It is critical to clarify the interpretation of single-person vs. interaction-focused metrics here: MPJPE, PA. and VPE focus solely on per-person reconstruction accuracy. Due to root alignment in their computation, they cannot capture errors in positioning or root jitter, which are critical for evaluating interaction quality. This inherent limitation explains why our method shows only marginal changes in MPJPE and VPE. By contrast, the substantial gains in RE, GE, and Int. directly validate that SocialMirror effectively addresses the challenges of severe mutual occlusions and disrupted spatial relationships, which are the primary pain points of monocular interaction reconstruction. In terms of inter-person penetration, this is not a metric that should be analyzed in isolation. This is because incorrectly placing two individuals engaged in close interaction far apart can also yield a small Pen. value. Notably, we expect slight model penetration to occur in close-interaction actions (e.g., hugging), where, for instance, one's palm may slightly intersect with the other person's body. To evaluate whether a method can minimize penetration while correctly inferring the relative positions of individuals, the three metrics—RE, Int. and Pen.—should be analyzed comprehensively. Specifically, our method maintains low values for RE, Int. and Pen. simultaneously. This demonstrates that it not only effectively captures the relative spatial relationships between individuals but also minimizes model penetration, thus ensuring the rationality of the interaction reconstruction results.

**Generalization Evaluation.** We further assess SocialMirror's generalization on the unseen Harmony4D dataset(Khirodkar et al. (2024))—without any fine-tuning—in Table 2, SocialMirror achieves superior results, especially on interaction metrics, validating its strong spatial relationship modeling. Figure 1 and Figure 4 illustrates qualitative reconstructions on in-the-wild interaction scenes, which the inputs are arbitrary video from the internet, demonstrating correct recovery of human contact and pose even under severe occlusion. These results confirm our framework's strong generalization across diverse datasets and real-world scenarios.

**VLM Annotation User Study.** Semantic information plays a vital role in our framework by complementing motion reconstruction. To assess the quality of interaction descriptions generated by the VLM Annotator, we conducted a user study with 20 participants who evaluated annotations from 40 randomly selected video sequences. Participants rated the alignment between VLM-generated texts and videos, as well as the accuracy of identified contact pairs, on a 5-point scale (1 indicating the generated description is completely irrelevant or incorrect which is below expectation and 5 indicating the description is exceptionally accurate and detailed which is exceeding expectation). The VLM Annotator achieved an average score of 3.3, where 3-score measuring the annotation is comparable with human annotation, demonstrating robust annotation quality and strong generalization across diverse interaction scenarios, with performance approaching human-level understanding.

## 4.4 ABLATION STUDY

**Ablation studies on each module** We conducted ablation studies to evaluate the impact of different modules in Table 3. Introducing the Semantic-Guided Motion Infiller module leads to a notable performance improvement, particularly in reducing GE and RE. The semantic information integration enables the network to preserve critical visual features while incorporating textual descriptions, leading to more accurate recovery of interactive motions and spatial relationships. Introducing the Temporal Refiner without factorized guidance or contact mask improves motion smoothness but degrades RE, GE and Int., confirming that temporal smoothing alone fails to resolve spatial inaccuracies. The Geometry Optimizer alone yields only modest improvements. In contrast, consistent improvements emerge when factorized guidance is employed: decoupling rotation and translation parameters allows independent tuning of hyperparameters for each and thus leads to superior convergence and overall reconstruction quality. Adding a contact mask further reduces RE, GE, and Int.. It primarily refines local details, such as hand–contact interactions. These fine-grained adjustments—typically centimeter-scale in localized regions—often manifest as subtle metric improvements that may not appear pronounced numerically. When all modules are combined, the full model achieves optimal performance, with the most significant gains observed in interaction-related metrics and motion smoothness. This validates the synergistic effect of each module. Notably, MPJPE, PA. and VPE metrics remain relatively stable across configurations, suggesting the model prioritizes global motion realism over joint-level precision—a trade-off favorable for visually realistic reconstructions.

**Comparison of various occlusion severities.** To further explore the effectiveness of our method on different occlusion-level cases, we quantified occlusion severity by computing intersection-over-union (IoU) between bounding boxes, partitioning the test set into five subsets representing distinct occlusion levels. As shown in Table 4, our approach achieves comparable results to CloseInt in scenarios without occlusion and consistently outperforms under partial and moderate occlusion (IoU between 0.25 and 0.75). With semantic and geometric guidance, the model reconstructs plausible poses by leveraging VLM-generated textual descriptions when visual cues are lacking, while also reducing mesh penetration and improving contact dynamics. Therefore, we obtain more naturally approximate real-world human interactions under challenging occlusion conditions.

## 5 CONCLUSION

We present SocialMirror, a diffusion-based method that integrates semantic cues and geometric constraints to address the challenges of monocular human mesh reconstruction in close-interaction scenarios. The Semantic-Guided Motion Infiller leverages vision-language descriptions to reconstruct occluded region and resolve pose ambiguities. Geometry Optimizer and Temporal Motion Refiner enforce 3D joint consistency and temporal consistency, enhancing spatial plausibility and natural contact dynamics. Extensive evaluations demonstrate that SocialMirror delivers realistic, semantically enriched reconstructions across various datasets and in-the-wild scenarios.

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

# A    SUPPLEMENTARY MATERIAL

We present additional details of implementation, including model setup, dataset processing, diffusion process modification and two-branch network architecture, as well as VLM annotator details with prompting examples in Sec.A.1. Additional experiments are provided in Sec.A.2, including ablation study on motion embedding layer design in Geometry Optimizer, performance breakdowns across Hi4D's action categories, cross-dataset results and in-the-wild visualizations. Section A.3 includes analyses of VLM limitations and failure cases, while also exploring the role of semantic information in limited-contact scenarios and outlining the framework's current limitations. The use of Large Language Models are declared in Sec.A.4

## A.1    ADDTIONAL DETAILS

### A.1.1    IMPLEMENT DETAILS

Our model was implemented using PyTorch and trained on an NVIDIA RTX 3090 GPU. The batch size was set to 32 for the Semantic-Guided Motion Infiller and 64 for the Geometry Optimizer. We employed the AdamW optimizer with CyclicLRWithRestarts, where the learning rate was initially set to 0.0001, with parameters restart_period=10, t_mult=2, and a "cosine" policy.

In the Motion Infiller and Motion Refiner, the dimension of human motion followed CloseInt(Huang et al. (2024)) with D = 157. For the Geometry Optimizer, we utilized 24 SMPL joints to represent human motion, resulting in a human motion dimension of D' = 24 × 3. The text feature dimension $F_{text}$, encoded from CLIP(Radford et al. (2021)), was 256.

For dataset implementation, original long motion sequences were divided into shorter clips with a length of L = 16 frames. Each clip was annotated with a corresponding text description using our LLM annotation module. For 3DPW, we established a new benchmark by selecting sequences involving two subjects: sequences captured in courtyard environments were used for training, and those captured in downtown settings were used for testing.

For multi-person scenes, we automatically detect and track individuals to obtain their bounding boxes and select the pair with the closest spatial proximity as the primary subjects. The original image is then cropped according to their bounding boxes, centering the region of interest to minimize background distractions and ensure the VLM focuses exclusively on the targets.

### A.1.2    DIFFUSION WITH INITIAL DISTRIBUTIONS

In prior approaches to diffusion-based pose estimation (Feng et al. (2023); Rommel et al. (2023)), time-dependent Gaussian noise sampled from $\mathcal{N}(0, I)$ is incrementally injected into ground-truth motion sequences $\hat{x}_0$ through the forward process:

$$q(\mathbf{x}_t \mid \hat{\mathbf{x}}_0) = \sqrt{\hat{\alpha}_t}\hat{\mathbf{x}}_0 + \sqrt{1 - \hat{\alpha}_t}\epsilon, \epsilon \sim \mathcal{N}(0, 1) \tag{2}$$

where $\alpha_t$ denotes a constant hyper-parameter(Nichol & Dhariwal (2021)), and $\hat{\alpha}_t = \prod_{i=0}^{t} \alpha_i$. It was observed that $x_t$ follows a standard Gaussian distribution, and the early iterative steps provide limited meaningful information for human motion dynamics. Additionally, the results should fully account for the initial prediction consistent with image characteristics.

To address these issues, we propose modifying the forward diffusion process to align with the initial distributions:

$$q(x_t|\hat{x}_0) = x + \sqrt{\hat{\alpha}_t}(\hat{x}_0 - x) + \sqrt{1 - \hat{\alpha}_t}\epsilon, \quad \epsilon \sim \mathcal{N}(0, \sigma) \tag{3}$$

With this adjusted framework, a generative model is derived by reversing the diffusion process, starting from samples $x_t \sim \mathcal{N}(x, \sigma)$. The reverse process is defined as:

$$q(x_{t-1}|x_t, c) = \mathcal{N}\left(x_{t-1}; \mu_\alpha(x_t, c), \tilde{\beta}_t \sigma\right) \tag{4}$$

where $\mu_\alpha(x_t, c)$ represents the estimated mean from the diffusion model under condition $c$ at timestep $t-1$, and $\tilde{\beta}_t$ denotes the variance calculated using the hyperparameters $\beta_t$, $\hat{\alpha}_t$, and $\hat{\alpha}_{t-1}$.

### A.1.3 MODEL DETAILS

We employ a two-branch network architecture to model human interactions, where each branch processes the actions of one individual and information sharing occurs between the branches. Specifically, $x_a^t$ and $x_b^t$ are first processed through a motion embedding layer and sequence position encoding to generate initial hidden states $h_a^0$ and $h_b^0$. These states are then fed into a two-branch transformer network with shared weights, composed of $N$ transformer blocks. Within each block, self-attention (SA) and cross-attention (CA) mechanisms enable intra-agent and inter-agent information exchange, respectively. For the n-th transformer block in agent $a$'s branch where $n \in [1, N]$ :

The Self-Attention Block processes its own hidden state $h_a^{n-1}$ to capture intra-agent dependencies. The query $Q_a^{\text{SA}}$, key $K_a^{\text{SA}}$, and value $V_a^{\text{SA}}$ matrices are derived from $h_a^{n-1}$ as:

$$Q_a^{\text{SA}} = h_a^{n-1} W_Q^{\text{SA}}, \quad K_a^{\text{SA}} = h_a^{n-1} W_K^{\text{SA}}, \quad V_a^{\text{SA}} = h_a^{n-1} W_V^{\text{SA}} \tag{5}$$

where $W_Q^{\text{SA}}, W_K^{\text{SA}}, W_V^{\text{SA}}$ are trainable weights. The self-attention output is calculated as:

$$\text{SA}(h_a^{n-1}) = \text{Softmax}\left(\frac{Q_a^{\text{SA}}(K_a^{\text{SA}})^T}{\sqrt{C}}\right) V_a^{\text{SA}} \tag{6}$$

where $C$ is the number of channels in the attention layer. Then a Cross-Attention Block facilitates inter-agent information exchange. For agent a, the query matrix $Q_a^{\text{CA}}$ is derived from $h_a^{n-1}$, while the key $K_a^{\text{CA}}$ and value $V_a^{\text{CA}}$ matrices come from $h_b^{n-1}$:

$$Q_a^{\text{CA}} = h_a^{n-1} W_Q^{\text{CA}}, \quad K_a^{\text{CA}} = h_b^{n-1} W_K^{\text{CA}}, \quad V_a^{\text{CA}} = h_b^{n-1} W_V^{\text{CA}} \tag{7}$$

The cross-attention output for agent a is:

$$\text{CA}(h_a^{n-1}, h_b^{n-1}) = \text{Softmax}\left(\frac{Q_a^{\text{CA}}(K_a^{\text{CA}})^T}{\sqrt{C}}\right) V_a^{\text{CA}} \tag{8}$$

A symmetric calculation for agent b, $\text{SA}(h_b^{n-1}), \text{CA}(h_b^{n-1}, h_a^{n-1})$, swaps the roles of $h_a^{n-1}$ and $h_b^{n-1}$. The weight matrices $W_Q^{\text{SA}}, W_K^{\text{SA}}, W_V^{\text{SA}}$ and $W_Q^{\text{CA}}, W_K^{\text{CA}}, W_V^{\text{CA}}$ are shared across both branches. At the end of each block, the outputs of the SA and CA blocks are combined with residual connections and layer normalization, for agent a:

$$h_a^n = \text{LayerNorm}\left(h_a^{n-1} + \text{SA}(h_a^{n-1}) + \text{CA}(h_a^{n-1}, h_b^{n-1})\right) \tag{9}$$

This integrated hidden state $h_a^n$ is then fed into subsequent transformer layers. The weight-sharing symmetry ensures balanced processing of inter-agent interactions, reducing model parameters while improving generalization capabilities.

Controlnet is a trainable copy of the N transformer blocks of the diffusion model, they share common inputs: $h_a^0$, $h_b^0$, t, and $F_{\text{img}}$. Additionally, it incorporates text features $F_{\text{text}}$ encoded by CLIP. For each trained transformer block, the computation is defined as: $h_i = \mathcal{T}(h^{i-1}, F_{\text{img}}; \Theta)$, where $\Theta$ denotes the frozen training parameters of the block.

The trainable copy of the model connects to the original model via zero linear layers. The output of the controlled diffusion network is therefore:

$$h_i^c = \mathcal{T}(h^{i-1}, F_{\text{img}}; \Theta) + \mathcal{Z}\left(\mathcal{T}\left(x + \mathcal{Z}(F_{\text{text}}; \Theta_{z1}), F_{\text{img}}; \Theta_c\right); \Theta_{z2}\right) \tag{10}$$

Here $\mathcal{T}$ represents the original model block $\mathcal{Z}$ denotes the zero linear layers. During the initial state of training, the zero linear layers produce zero outputs, ensuring the original model's stable output as $h_i^c = \mathcal{T}(h_{i-1}^c; \Theta)$. As training progresses, the parameters of the zero linear layers learn to gradually inject conditional signals into the model.

### A.1.4 VLM ANNOTATOR DETAILS

We further provide the details of LLM Annotation in Tab. 5. We also provide several generated textual descriptions and contact pairs in our visualization results (Fig. 5), illustrating that the texts are well-aligned with the corresponding image, providing semantic guidance for human mesh reconstruction.

Table 5: Detailed prompting example for VLM Annotator.

| Prompting Example |
| --- |
| Given the image sequence of two human interaction, generate 0, 1 or more joint-joint contact pair(s) according to the following background information, rules, and examples. Joint-joint contact pair should exactly reflect the human interaction shown in the image sequence. |
| [Start of background Information] |
| Human has JOINTS: ['pelvis', 'left_hip', 'right_hip', 'left_knee', 'right_knee', 'left_ankle', 'right_ankle', 'left_foot', 'right_foot', 'neck', 'left_collar', 'right_collar', 'head', 'left_shoulder', 'right_shoulder', 'left_elbow', 'right_elbow', 'left_wrist', 'right_wrist' ]. |
| [End of background Information] |
| [Start of rules] |
| 1.Each joint-joint pair should be formatted into {JOINT, JOINT, TIME-STEP, TIME-STEP}. JOINT should be replaced by JOINT in the background information. IMPORTANT: The first JOINT belongs to person 1, and the second JOINT belongs to person 2. Each joint-joint pair represents a contact of a joint of person 1 and a joint of person 2. The first TIME-STEP is the start frame number of contact, and the second TIME-STEP is the end frame number of contact. |
| 2.Use one sentence to describe what action person 1 do and one sentence to describe what action person 2 do according to the image sequence. IMPORTANT: the sentence starts from 'text 1:' describing the action of person 1 from the perspective of person 1 and the sentence starts from 'text 2:' describing the action of person 2 from the perspective of person 2. Sentences should NOT contain words like 'person 1' or 'person 2', use 'a person' to refer to himself in the sentence and 'others' to refer to others. IMPORTANT: the sentence should be align with the joint-joint contact pair. IMPORTANT: the order of person 1 and person 2 should be the same in different joint-joint contact pair of the same image sequence. |
| 3.IMPORTANT: Do NOT add explanations for the joint-joint contact pair. |
| [End of rules] |
| [Start of an example] |
| [Start of sentences] |
| Text 1: a person dance with others holding his left hand with the other's right hand, puting his right hand on the other's waist, and his shoulder being touched. |
| Text 2: a person dance with other holding her right hand with the other's left hand, with her waist being embraced, placing her left hand on the other's shoulder. |
| [End of sentences ] |
| [Start of joint-joint contact pair(s)] |
| {left_wrist, right_wrist, 11, 15} |
| {right_wrist, left_hip, 14, 15} |
| {right_shoulder, left_wrist, 9, 15} |
| [End of joint-joint contact pair(s)] |
| [End of an example] |

## A.2 ADDTIONAL EXPERIMENTS

### A.2.1 GEOMETRY OPTIMIZER

Geometry Optimizer focuses on processing 3D joint positions to provide geometric guidance information. To validate the effectiveness of our encoding layer design for the auxiliary model, we conducted an ablation study by implementing the motion embedding layer with either STGCN or a Linear layer. The results are presented in Tab. 6.

Table 6: Ablation studies on the impact of motion embedding layer.

| Linear | STGCN | ↓R-MPJPE | ↓G-MPJPE | ↓Int. | ↓MPJPE | ↓PA-MPJPE |
| --- | --- | --- | --- | --- | --- | --- |
| ✓ | | 102.3 | 110.0 | 84.9 | 81.9 | 66.8 |
| | ✓ | **81.7** | **93.2** | **62.5** | **60.8** | **47.8** |

The Geometry Optimizer that uses STGCN to encode 3D joint positions exhibits higher accuracy than the one using Linear. It successfully captures the 3D positional relationships of interacting humans and outperforms Motion Infiller in all metrics related solely to 3D joint positions. This indicates that it can effectively provide correct guidance information.

### A.2.2   ADDITIONAL EXPERIMENTS RESUTLTS ON HI4D

**Comparison on Various Actions.** We further split the Hi4D dataset into subset based on action labels to validate our effectiveness on various action categories. Tab. 7 presents our method's improvements over CloseInt across different subsets. Notably, our approach achieves the most significant gains on actions such as handshake, high-five, and kiss. In these actions, human behavioral patterns are relatively uniform, and occlusion levels are moderate. The model synthesizes plausible poses by integrating textual descriptions generated by VLM Annotator, while simultaneously mitigating mesh interpenetration issues and refining contact dynamics. However, the method shows smaller gains on complex actions such as dancing and fighting. These activities involve intricate limb interactions and ambiguous joint-depth relationships, which can slightly undermine VLM annotation consistency and the precision of geometric guidance. Nonetheless, our method still outperforms the baseline.

Table 7: Comparison of CloseInt and our method, CloseInt/Ours (Improvement), across different actions on Hi4D.

| Action | handshake | highfive | kiss | dance | fight |
|---|---|---|---|---|---|
| ↓R-MPJPE | 78.0/65.8(20.0) | 60.5/53.5(19.2) | 81.7/67.7(**20.9**) | 96.4/87.6(9.4) | 110.3/100.3(8.2) |
| ↓G-MPJPE | 93.2/72.5(23.8) | 70.9/84.9(**27.8**) | 98.9/79.7(19.6) | 109.2/97.9(9.4) | 131.4/120.1(6.5) |
| ↓Int. | 36.9/31.1(15.6) | 26.7/25.1(5.9) | 33.3/23.0(**63.9**) | 39.9/32.4(18.8) | 46.7/41.1(11.9) |
| ↓Pen. | 194.8/71.9(63.1) | 107.4/50.1(53.3) | 15409.3/5570.1(63.9) | 5477.1/2455.0(55.2) | 636.6/226.2(**64.5**) |

### A.2.3   CROSS DATASET EVALUATION

We also report both intra-domain and cross-domain results. SocialMirror outperforms prior methods in all settings. In our experiments, we observed that when not trained on the dataset, CloseInt may erroneously separate characters that should be in close contact. This results in the absence of even minor intended penetrations (e.g., slight mesh intersection between a palm and another person), leading to a relatively low penetration error—though this is not indicative of a good reconstruction outcome. After training on the dataset, CloseInt's errors in character placement are reduced, but it correspondingly exhibits more interpenetration, which explains why the penetration loss increases post-training. Our method, in both scenarios, produces more accurate relative positions of characters (as reflected in RE and Int.) while ensuring less interpenetration, demonstrating the positive effect of the proposed method in reducing interpenetration.

### A.2.4   RESULTS ON HARMONY4D

For completeness, we also conducted training experiments on the Harmony4D dataset, which further confirms the effectiveness of our approach. Specifically, our method achieves significant improvements in interaction-related metrics: it yields increases of 8.2%, 3.5%, and 3.2% in RE, GE, and Int., respectively. Meanwhile, it maintains nearly unchanged performance on single-person reconstruction metrics (i.e., MPJPE, PA., and VPE). This result demonstrates the robust capability of our method in capturing human interaction relationships.

### A.2.5   ADDITIONAL VISUALIZATION RESULTS

We present additional reconstruction results on in-the-wild data in Fig.5, include a detailed comparison and demonstration of the results in our accompanying video.

Table 8: Cross Dataset Evaluation on Hi4D and 3DPW.

| Method | Hi4D | | | | | | | 3DPW | | | | | | |
|---|---|---|---|---|---|---|---|---|---|---|---|---|---|---|
| | ↓RE | ↓GE | ↓Int. | ↓Pen. | ↓MPJPE | ↓PA. | ↓VPE | ↓RE | ↓GE | ↓Int. | ↓Pen. | ↓MPJPE | ↓PA. | ↓VPE |
| CloseInt | 99.0 | 114.9 | 81.4 | 3947.6 | 63.1 | **47.5** | **76.4** | 135.7 | **159.1** | 95.5 | 342.7 | 79.9 | 52.9 | 95.1 |
| Ours | **83.6** | **95.2** | **68.5** | **2380.5** | **62.2** | **47.5** | 79.3 | **104.8** | 162.7 | **89.9** | **109.7** | **65.1** | **49.0** | **79.7** |
| CloseInt(Eval. Only) | 181.1 | 232.1 | 182.7 | **1973.8** | 109.1 | **62.5** | 132.0 | 194.4 | 340.2 | 128.4 | **101.6** | 88.6 | 63.6 | 110.7 |
| Ours(Eval. Only) | **165.2** | **184.1** | **153.0** | 2380.3 | **105.2** | 63.6 | **129.4** | **174.7** | **307.4** | **125.4** | 109.7 | **87.5** | **63.3** | **109.8** |

Table 9: Comparisons on Harmony4D.

| | ↓RE | ↓GE | ↓Int. | ↓Pen. | ↓MPJPE | ↓PA. | ↓VPE |
|---|---|---|---|---|---|---|---|
| CloseInt | 134.8 | 297.5 | 182.5 | 482.6 | 70.2 | 38.6 | 82.6 |
| Ours | **123.8** | **287.2** | **176.8** | **480.3** | **69.8** | 39.7 | **80.8** |

## A.3 DISCUSSIONS

### A.3.1 RECONSTRUCTION UNDER VLM LIMITATIONS

Based on our user study, the text descriptions generated by the VLM are, on average, superior to those produced by human annotators. As shown in Fig.5, VLM annotations can capture not only macroscopic actions but also fine-grained contact relationships between specific joints (e.g., "A person leads the dance, extending his left arm to hold the other's right hand and guiding her movements with his right hand on her back"), whereas a human annotator might simply describe it as "two people dancing ballroom dance". While VLM Annotator demonstrates satisfactory performance in describing human interaction under most circumstances, its accuracy tends to decline when confronted with complex limb interactions, affecting the precision of both textual descriptions and contact pair annotations. By prioritizing visual feature extraction over textual inputs, our proposed method maintains reconstruction fidelity even when text-image alignment is compromised. As illustrated in Fig.6, despite VLM Annotator's failure to correctly identify the human action, our approach successfully reconstructs accurate motion patterns by leveraging visual information.

### A.3.2 FAILURE CASES

It should be acknowledged that our methodology exhibits limitations in scenarios involving prolonged and severe occlusions. Fig.7 exemplifies such a challenging case where both visual and semantic information are critically compromised. The inaccurate textual annotations and contact pair predictions generated by VLM Annotator in this context lead to erroneous guidance signals, resulting in substantial reconstruction deviations. This observation underscores the necessity of complementary mechanisms to handle extreme occlusion scenarios in future work.

### A.3.3 THE EFFECT OF SEMANTIC INFORMATION ON LIMITED CONTACT SCENARIOS

Even when contact is absent, the VLM can produce high-level scene descriptions (e.g., two people stand and face each other), which are encoded as semantic features. These provide contextual cues about interaction and spatial layout beyond direct contact information. In addition, our approach does not rely solely on contact labels. The semantic features guide the Motion Infiller to infer plausible poses for ambiguous regions, and the Temporal Refiner and geometric constraints based on 3D joint prediction from the Auxiliary Module ensure motion smoothness and spatial plausibility. Table 5 further shows interaction metrics gains even for non-close interactions with mild occlusion and few contacts.

### A.3.4 LIMITATIONS

For future work, a promising direction to improve the reliability and overall robustness of the semantic guide of the method is to explore strategies to mitigate the inherent inability of the VLM, including, but not limited to, introducing the confidence scores of the VLM Annotator and the weighting of the guidance, and the development of effective mechanisms for verifying semantic and visual consistency.

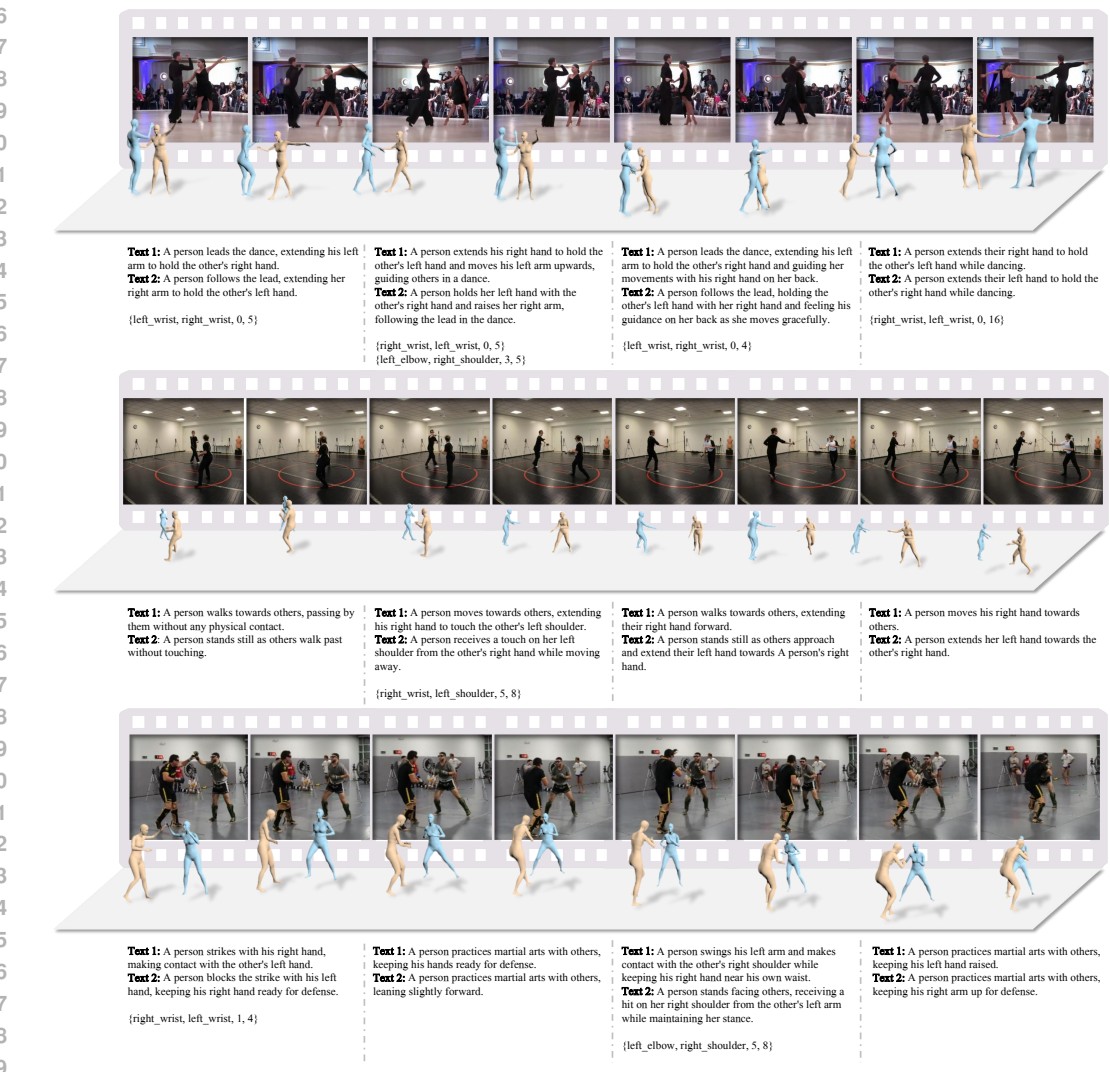

Figure 5: Visualization results on in-the-wild data.

It should also be noted that the proposed method is currently limited to reconstructing interactions between two individuals. For reconstructing interactions involving more participants, further improvements to the network architecture are required.

## A.4 THE USE OF LARGE LANGUAGE MODELS (LLMS)

It is hereby declared that in this paper, Vision-Language Models (VLMs) are primarily employed as a VLM Annotator, whose primary function is to generate textual descriptions of human interactions within image sequences and spatio-temporal joint contact pairs. Additionally, LLMs are utilized solely for textual polishing and grammatical correction, the overall research approach, core insights, reasoning processes, and final conclusions presented in this paper constitute the independent outcomes of the authors' original intellectual work and research endeavors. All instances involving the generation of content through VLMs and LLMs are explicitly documented, with detailed accounts of their application methods and scenarios.

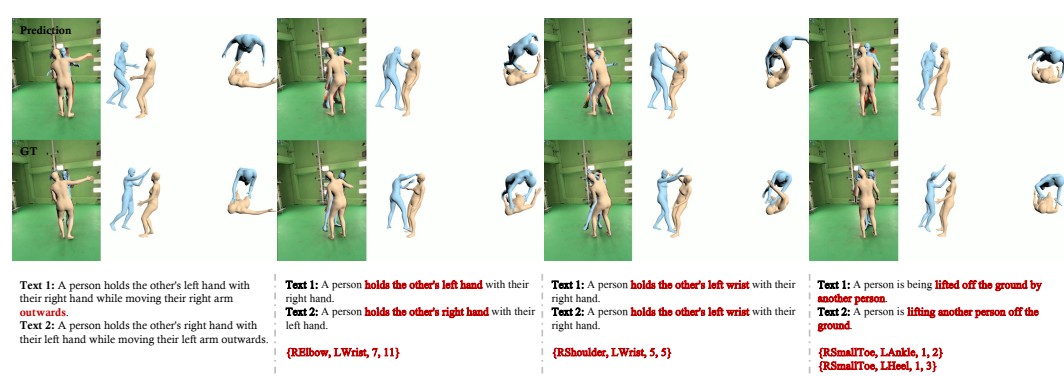

**Text 1:** A person holds the other's left hand with their right hand while moving their right arm outwards.
**Text 2:** A person holds the other's right hand with their left hand while moving their left arm outwards.

**Text 1:** A person **holds the other's left hand** with their right hand.
**Text 2:** A person **holds the other's right hand** with their left hand.

{RElbow, LWrist, 7, 11}

**Text 1:** A person **holds the other's left wrist** with their right hand.
**Text 2:** A person **holds the other's left wrist** with their right hand.

{RShoulder, LWrist, 5, 5}

**Text 1:** A person is being **lifted off the ground by another person**.
**Text 2:** A person is **lifting another person off the ground**.

{RSmallToe, LAnkle, 1, 2}
{RSmallToe, LHeel, 1, 3}

Figure 6: VLM Annotator failed to describe human interaction.

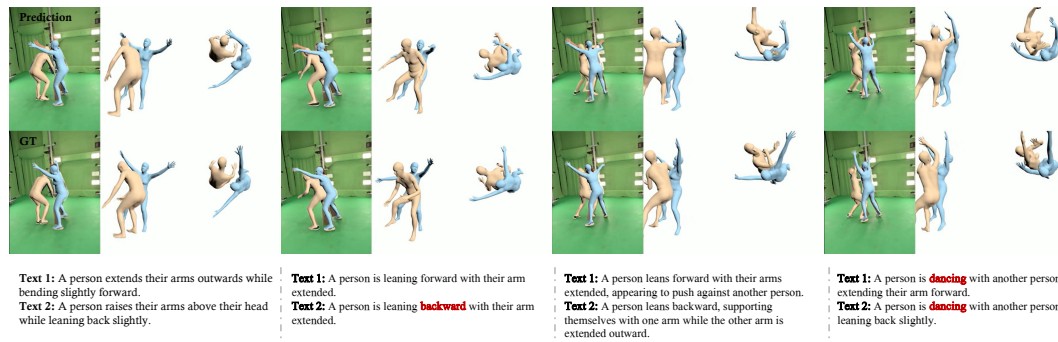

**Text 1:** A person extends their arms outwards while bending slightly forward.
**Text 2:** A person raises their arms above their head while leaning back slightly.

**Text 1:** A person is leaning forward with their arm extended.
**Text 2:** A person is leaning **backward** with their arm extended.

**Text 1:** A person leans forward with their arms extended, appearing to push against another person.
**Text 2:** A person leans backward, supporting themselves with one arm while the other arm is extended outward.

**Text 1:** A person is **dancing** with another person, extending their arm forward.
**Text 2:** A person is **dancing** with another person, leaning back slightly.

Figure 7: Challenging case which involves prolonged and severe occlusions.

