# OpenReview forum: "SocialMirror: Reconstructing 3D Human Interaction Behaviors from Monocular Videos"
_ICLR.cc/2026/Conference — ICLR 2026 Conference Withdrawn Submission_

### Official Review · Reviewer_Eca9 · 2025-10-31

**Soundness:** 3
**Presentation:** 2
**Contribution:** 2
**Rating:** 2
**Confidence:** 5

**Summary:**

This paper proposes a framework for multi-person 3D mesh estimation, leveraging VLMs for semantic-guided refinement. The authors present a relatively complete system with module-wise ablation studies, robustness evaluation, and cross-dataset experiments. The main idea of using VLMs to provide semantic guidance in multi-person mesh estimation is intuitive, but there are several concerns regarding novelty, experimental validation, and presentation.

**Strengths:**

1. The method is easy to understand and the overall framework is straightforward to follow.

2. Introducing a VLM for semantic-guided refinement is an interesting idea; intuitively, it could help improve mesh estimation in challenging cases.

3. The paper includes module ablation studies, robustness checks, and cross-dataset experiments, making the evaluation relatively comprehensive.

**Weaknesses:**

1. Writing and presentation issues:

   - Numerous typos and basic grammar mistakes (e.g., missing spaces around parentheses) make reading difficult.

   - Writing is sometimes non-standard; tables are unclear and inconsistent, e.g., bold in Table 7 is confusing. Also in Table 3, the second column’s module is unclear; the second-to-last row (68.5) should be bolded.

   - Section 4.3 has large blocks of text without paragraphs, which hinders readability.

   - Figure 1 and method nickname: It is unclear what Figure 1 aims to convey. The term SocialMirror is never explained, and the connection to “mirror” is unclear.

2. Impact of contributions is concerned:

    - While the VLM module is intended to provide semantic guidance, ablation results show that it sometimes reduces performance on certain metrics.

    - Improvements from other modules are relatively small and occasionally worsen metrics, raising concerns about the overall effectiveness.

    - Upon checking the supplementary video, there are noticeable errors in multi-person mesh estimation (e.g., around 30 seconds in one video), which casts doubt on the claimed performance.

    - The VLM Annotation User Study reported an average score of 3.3/5, which is only moderate and may reflect why some ablation metrics decrease when using VLM.

    - Table 4 shows several metrics degrading in later entries; the explanation in line 417 is not very convincing.

**Questions:**

Could the authors clarify the meaning of SocialMirror and what Figure 1 is intended to illustrate?

---

### Official Review · Reviewer_9YpQ · 2025-11-02

**Soundness:** 3
**Presentation:** 3
**Contribution:** 3
**Rating:** 6
**Confidence:** 3

**Summary:**

The paper proposes a diffusion framework for reconstructing human behavior in two-person interaction scenarios with mutual occlusion, where traditional HMR is difficult to apply due to the lack of direct visual cues for the occluded person. The method is guided by high-level semantic knowledge from a VLM as well as low-level geometry, contact, and temporal smoothness constraints. First, a VLM is fine-tuned to produce semantic captions for the interaction motion of two people, as well as explicit contact labels. The semantic captions are encoded by CLIP and concatenated with HMR2.0 features. A coarse SMPL pose is also predicted from HMR2.0, and further refined via temporal diffusion, where the CLIP+HMR2.0 embedding is used as ControlNet-style diffusion conditioning. A graph convolutional network is introduced to predict 3D joint locations from SMPL parameters. Finally, InterControl is used to optimize the original SMPL motions to minimize contact and inter-penetration constraints.

**Strengths:**

- Reconstructing human motions in close two-person human-interaction scenarios is an important and challenging problem.
- The method produces state-of-the-art results on Hi3D, 3DPW, and Harmony4D.
- The proposed design is effective according to ablation study

**Weaknesses:**

- The method is designed for two-person interaction scenarios, but it is unclear whether it easily scales to multi-person interaction scenarios with mutual occlusions, such as those in 3DPW.

**Questions:**

- My understanding is the auxiliary model takes SMPL poses as input, and produces 3D joint locations as output. Why not just use the original SMPL inference procedure to obtain 3D joint locations?
- The second column in Tab. 3 is labeled with a dash (between "Semantic-guided motion filler" and "Temporal refiner factorized guidance") - what does it represent?
- The inter-person penetration volume does not seem to match between Tab. 1 (ours = 2380.5) and Tab. 4 (max(ours) = 406.3). Are these evaluated on the same Hi4D test set? Is there any intuitive explanation for the large discrepancy?

---

### Official Review · Reviewer_xv5C · 2025-11-03

**Soundness:** 2
**Presentation:** 2
**Contribution:** 2
**Rating:** 2
**Confidence:** 5

**Summary:**

- The paper focuses on mesh recovery from monocular videos of two-subject interactions.
- The method uses text-guided motion infilling via a diffusion model, then refines mesh trajectories with auxiliary and collision losses to produce realistic, image-consistent human meshes.
- Ablations highlight the importance of the semantic-guided motion infiller and geometry optimizer, crucial for improving metrics, especially RE and GE.
- Evaluations span three datasets: Hi4D, 3DPW, and Harmony4D.
- Key baselines: BUDDI, Human4D, CloseInt.
- Quantitative results show that SocialMirror consistently outperforms baselines across datasets and generalizes to sequences outside the training distribution (e.g., Harmony4D).

**Strengths:**

- The paper is well-organized and easy to follow.
- The use of VLM-derived, text-guided semantics for mesh recovery is novel. While its viability warrants further study, it opens directions for VLMs that predict mesh parameters end-to-end directly from video.
- Evaluations span multiple datasets and scenarios. The supplementary qualitative results are insightful and show SocialMirror consistently outperforming Human4D and CloseInt across settings.

**Weaknesses:**

- Motivation of using a VLM for close-interaction reasoning: The method uses a VLM to summarize the interaction video, which then conditions the Interactive Diffuser to generate mesh sequences. This pipeline can be brittle: the pixels-to-text step is inherently lossy, and capturing fine-grained human motion in text is challenging. Although the Interactive Diffuser also takes images as input, VLM-derived conditioning may introduce bias or errors and degrade performance. It would be helpful to ablate SocialMirror’s robustness with respect to the accuracy of the semantic-text guidance.

- Restricted to two people interaction: The methodology appears restricted to two-person interactions and seemingly cannot handle single-person sequences or scenes with more than two people (please correct me if I am wrong). This design choice is limiting and constrains real-world applicability. Notably, this is also a key limitation of related works in this area (e.g., BUDDI, ClostInt).

- Qualitative Comparisons: The key request here is to test the generalization ability of SocialMirror beyond Hi4D (train set) and 3DPW like setting. I appreciate the Harmony4D qualitative results; however, comparable baselines on these videos (e.g., BUDDI, CloseInt) are missing. Including them would demonstrate out-of-distribution robustness and help rule out overfitting.

- Runtime analysis: Most related baselines are feed-forward with optional lightweight optimization. In contrast, the proposed method includes heavier components (e.g., the VLM and diffusion-based motion infilling) that may increase inference latency. Additionally, given the number of modules, the manuscript does not clearly separate train-time and inference-time modules. A runtime/throughput and memory breakdown with baselines like Human4D, BUDDI and CloseInt would be very helpful here.

**Questions:**

My key questions are centered around the weaknesses mentioned above:
1. Performance degradation study with respect to accuracy of the VLM summary.
2. Extending to single person and beyond two person scenarios.
3. Qualitative comparison on out-of-domain videos.
4. Runtime analysis wrt baselines.

---

### Note · Authors · 2025-11-13

I have read and agree with the venue's withdrawal policy on behalf of myself and my co-authors.